# Experimental and Simulation Study of the Latest HFC/HFO and Blend of Refrigerants in Vapour Compression Refrigeration System as an Alternative of R134a

Uma Shankar Prasad [1,2,*], Radhey Shyam Mishra [3], Ranadip Kumar Das [1] and Hargovind Soni [4,*]

1   IIT(ISM), Dhanbad 826004, India
2   PIEMR, Indore 452010, India
3   Department of Mechanical Engineering, Delhi Technological University, New Delhi 110006, India
4   Department of Mechanical Engineering, National Institute of Technology Delhi, Delhi 110036, India
*   Correspondence: dp16dp.16dp000005@mech.iitism.ac.in or ushankar@piemr.edu.in (U.S.P.); hargovinds@nitdelhi.ac.in (H.S.)

**Abstract:** Experimental and simulation investigation of the performance and characteristics of different refrigerants and blends of refrigerants is carried out to replace the existing refrigerant R134a for a vapour compression refrigeration system. The performance of VCRS systems was improved by several researchers by introducing the concept of mixing the family of refrigerants with low GWP in the working circuit. This research paper presents the performance results of different refrigerants and blends of refrigerants that can replace the R134a it is also an attempt to cover the mechanism and possible combination of different blends of refrigerants to improve the effectiveness as well as efficiency of the refrigeration system. Detailed analysis of different parameters of heat transfer and predictions of low-GWP refrigerants, including the HFO (hydro fluoro-olefin) class and the HC (hydrocarbon) class through energy and exergy analysis of commercial refrigerants such as R134a is performed. Results are obtained by using an experimental test rig and the input parameters of the experiments are kept the same with the simulation software (CYCLE_D-HX 2.0) and validated with the results to replace R134a.

**Keywords:** hydrofluoroolefins; hydrofluorocarbons; refrigeration system; energy technology; environmentally friendly; alternative refrigerants; low GWP

## 1. Introduction

The process of maintaining and achieving the temperature below the surrounding is known as refrigeration. The main aim is to lower the temperature of the product or space to the desired one. Maxwell et al. [1] discussed the history of refrigeration with the basic concept of modern refrigeration systems using ammonia, carbon dioxide and aqua-ammonia as a refrigerant. Several researchers investigated and found that efficiency can be enhanced by the blending of refrigerants with different proportions. During several investigations, it was noted that these blended refrigerants have several novel properties that make them very useful in various types of applications such as fuel cells, heat sinks, heat exchangers, heat transfer, hybrid engines, pharmaceutical processes, microelectronics, grinding, machining, the cooling system of different components, chillers, domestic refrigerators, solar applications, greenhouse applications. They can improve the thermal conductivity of the base fluid as well as it was noted that convective heat transfer is also significantly improved. According to an international agreement (Montreal Protocol, 1987), the use of CFC halogenated refrigerants such as R-11, R-12, R-113, R-114, and R-502 that have high ODP have been phased out. The most widely used refrigerants R-11 and R-12 are replaced because of their long-term greenhouse effects with R134a. HFC -134a is a non-flammable hydrofluorocarbon and is the best alternative to R-12 CFC

refrigerant, but due to high GWP of 1430, insolubility and incompatibility with organic mineral oils, the Environmental Protection Agency (EPA) listed R134a in the category of unacceptable refrigerant. In this paper, the behaviour of suitable refrigerants and blended mixture of refrigerants is investigated experimentally and simulation (CYCLE_D-HX 2.0) results of software, to show the new possibilities of utilisation of different blended refrigerants in the field of refrigeration and HVAC systems to replace R134a refrigerant. The mixing and use of different blended refrigerants were started in the third generation of refrigeration in the 1990s, i.e., HFC refrigerants. Mixing of some refrigerants reduces the flammability and toxicity, it was also observed that some refrigerants have very high pressure and by mixing of refrigerants the working pressure is reduced. The normal boiling pressure is also affected and changed by the blending process resulting in a change in the cooling capacity of the system. Spauschus et al. [2] discussed the adoption of R134a as a replacement for R12 in compressor and refrigeration systems for commercial purposes to comply with the Montreal Protocol. The desired physical and chemical properties of R134a were reviewed, including its behaviour with lubricants. However, a complete assessment, including refrigerant process development, toxicological validation, lubricant development, material screening tests, and refrigeration product engineering, would be required before commercialisation. Butterworth et al. [3] tested propane and a mixture of propane and isobutene and found that the mixture can be used as a "drop-in" replacement for R12, with improved COP. Havelsky et al. [4] conducted a study to compare different refrigerants as a replacement for R12 in terms of energy efficiency, COP, and TEWI. They compared R134a, R401A, R409A, R22, and a mixture of R12 and R134a. They concluded that R134a, R401A, and R409A showed better COP results than R12 and reduced TEWI levels. Domanski et al. [5] compared the performance of R134a and $CO_2$ refrigerants using semi-theoretical cycle models CYCLE-11.UA and CYCLE-11.UA-$CO_2$ and found R134a has better COP results than $CO_2$. Sekhar et al. [6] investigated the replacement of CFC12 refrigerant in a household refrigerator with an eco-friendly blended refrigerant mixture of HFC134a/HC290/HC600a and found refrigerant mixture is a better replacement for CFC12. Gigiel et al. [7] performed various tests, including a pressure test, scratch test, leak test for protected circuits, leak test for unprotected circuits, leak test for external joints, and measurement of concentration with R600a. Hosoz et al. [8] compared the performance of a single stage refrigeration system and a cascade refrigeration system, both using R134a as the refrigerant. The cascade system consumed more compressor power overall due to the second compressor in the high-temperature loop. The volumetric efficiency of the single stage system was lower than the low temperature section of the cascade system. The overall COP of the cascade system was low due to the second compressor in the high-temperature loop. Gang et al. [9] conducted a performance analysis on a domestic refrigerator using different ratios of HFC152a/HFC125 refrigerant mixture and found a mass fraction of 0.85 of HFC152a with prior used refrigerant CFC12 yielded the same results. Chimres et al. [10] investigated the performance of refrigerants R290 (propane), R600 (butane), and R600a (isobutane) in a 239 L capacity refrigerator with a 53 L freezer and 100-watt power consumption and the best results were obtained with a mixture of 60% propane and 40% butane. Fatouh et al. [11] analysed and concluded that hydrocarbon refrigerants have zero ODP and GWP and the best results were obtained with a mixture of 60% propane with n-butane and iso-butane. The use of 70% propane in the mixture increased the volumetric efficiency by 15.5% and the coefficient of performance can be improved by 2.3% by using 60% propane. Ding et al. [12] conducted a comprehensive review of the use of simulation models for vapour compression refrigeration systems to optimise their design and predict performance. Jwo et al. [13] performed experiments with R134a working refrigerant replaced by R290 and R-600a hydrocarbons of 50:50 ratios and found that the refrigerating effect is improved. The total energy consumed by 4.4% also the mass of refrigerant for charging the system is reduced by 40%. Mohanraj et al. [14] conducted an experimental study using a mixture of R290 and R600a hydrocarbon refrigerants under different ambient temperatures ranging from 24 to 43 °C resulted in a reduction in power

consumption of the refrigerator. Padilla et al. [15] conducted an analysis of energy balance on a household domestic refrigerator using R12 and found that the overall performance of the refrigerator using the zeotropic mixture was better than that of R12, provided that the evaporator temperature was maintained between 15 °C and −10 °C. Agarwal and Srivastava et al. [16] conducted experiments to test the performance of the system with different eco-friendly hydrocarbon refrigerants and observed that these refrigerants resulted in a desired reduction of CFC emissions. Wongwises et al. [17] compared the performance of CFC12, CFC22, and HFC134a refrigerants with alternative refrigerants HC290, HC1270, HC600, and HC600a at different ratios. Tiwari et al. [18] conducted experiments using refrigerants R404a and R134a and found R134a consumed less energy but in all working conditions, the performance of R404a was significantly better than R134a. Bolaji et al. [19] investigated the performance of HFC refrigerants R32, R134a, and R512a and found that R32 had a low COP and very high operating pressure. The performance of R152a and R134a was similar at different temperatures, but the COP of R152a was higher than both R32 and R134a. In addition, R152a had zero ODP and very low GWP. Liu et al. [20] conducted experimental investigations on two different vapour compression systems with the use of a mixed blend of R290 and R600 in the 20-cubic feet system resulted in 6% energy savings, the use of hydrocarbon blended mixtures in the 18-cubic feet system resulted in energy savings of up to 17.3%. Mishra et al. [21] performed numerical computations to analyse the thermal performance of a three-stage cascade vapour compression refrigeration system and analysis showed that R-600a refrigerant yielded the best system performance at an ultra-low temperature of −155 °C. Domanski et al. [22,23] presented the simulation software developed by the national institute of standards and technology (NIST) which imports the most accurate input parameters of thermos-physical properties of different fluids and fluid mixtures, as it has standard reference data and is validated by more than 200 countries. Ian H. Bell et al. [24] investigated the blend of the 23 best refrigerants of low GWP through simulation software REFPROP and CYCLE_D-HX to compare the results as a replacement for the R134a refrigerant. Domanski et al. [25,26] did an investigation on simulation tool CYCLE11 used for the preliminary evaluation of refrigerants and refrigerant mixtures in the vapor-compression cycle. The program is based on the Carnahan-Starling-DeSantes equation of state and assumes an isenthalpic expansion process. It includes a simple model of the compressor and considers heat exchange in the suction and liquid lines. Domanski et al. [27] developed a simulation model called CYCLE_D-HX to evaluate the transport properties and optimise heat exchange in heat exchangers. They conducted an experiment to evaluate the performance of R134a, R600a, and R-32 refrigerants and validated the data with the CYCLE_D-HX model (Figure 1). Gil et al. [28] investigated the efficiency of HFO/HCFO refrigerants in the ejector cooling cycle with three different levels of condensation and evaporation temperatures. The results showed that hydro-fluoro olefins, particularly HFO-1234zf and HFO-1234ze(E), can achieve high efficiency in the ejector cooling cycle. Adelrajafi et al. [29] developed a CSA-LSSVM model to predict the behaviour of a low GWP binary mixture of refrigerants, R-1234yf and R-1234ze(E). They compared their model's predictions to those of the PREOS and PC-SAFT models and found that their CSA-LSSVM model had better performance. Van Vu Nguyen et al. [30] investigated the performance of six refrigerants, R-1234ze(E), R-32, R-152a, R-290, R-600a, and R-1234yf, in terms of COP, operating pressure, and sensitivity of ejector geometry under different working conditions and found HFC R-152a and HFO R-1234ze(E) performed the best, R-600a was the most favourable, and R-1234yf was compatible with R-290. Emmi et al. [31] presented the configurations and monitoring data to study the behaviour of a two-stage heat pump operated with R-744 in the ejector system and secondary refrigerant R-1234ze is used in the high-temperature stage. The study aimed to find out the effective energy performance of the system. Taweekum et al. [32] analysed R-463A as an alternative to R-404 in a NIST vapour compression cycle model. Using CYCLE_D-HX software, they found that R-463A has a higher normal boiling point and a significantly lower GWP than R-404A. R-463A can also be used in high ambient temperature environments due

to its higher critical pressure and temperature. At low temperatures, R-463A has a 10% higher COP than R-404A. Gil et al. [33] proposed new three-component refrigerants with a 10% step in mass fraction, using a triangular design. The researchers found that the best performing mixture was R1234yf-R152a-RE170 with a weight share of 0.1/0.5/0.4. Andreas et al. [34] studied the two-phase condensation heat transfer process and pressure drop characteristics of R-513A and found that the pressure drop of R-513A was similar to R-1234yf and 10% lower than that of R134a at higher mass flux. However, the pressure drops of R-1234ze(E) were 20% higher compared to those of R134a at higher mass flux. Bharanitharan et.al [35] conducted a study on the hydrodynamics of an oscillating Stirling regenerator at various speeds and compared the experimental and numerical results. They found that the numerical results were able to predict the flow behaviour in the regenerator, and the Ergun correlation performed well at high flow rates. Kumar et al. [36] conducted a review on low GWP refrigerants such as R-1234ze(E), R-1234ze(Z), R-1234yf, R-513A, and R-450A as substitutes for R134a. They analysed the thermodynamic and transport properties of these refrigerants using experimental, numerical, and simulation studies. Nikitin et al. [37] conducted an investigation on the performance of a heat pump on the soil at different temperatures and variable depths. They used a mixture of R-41 and R-161 to understand the effect of ice thickness and snow cover by employing computational fluid dynamics and thermo-economic–environmental analysis on the cascade system. Deyni et al. [38] developed a model to evaluate six pairs of refrigerants for use in a cascade refrigeration system. The refrigerant pairs evaluated were R41-R161, R41-R1234yf, R41-R1234ze, R744-R161, R744-R1234yf, and R744-R1234ze. The results showed that R41-R161 and R41-R1234ze had the highest COP. Nikitin et al. [39] conducted a comparative study of energy, exergy, economic, and environmental analysis of the 10 coldest Russian cities using the Pareto front curve and found Saint Petersburg would benefit from using air-source heat pump (ASHP) systems, while Khabarovsk city would benefit from using ground-source heat pump (GSHP) systems. Dashtebayaz et al. [40] studied the efficiency of five HFC refrigerants on geothermal heat pumps to optimise system design, finding R-134a had the highest efficiency and R-125 the lowest. Dashtebayaz et al. [41] studied the use of an air source heat pump as a waste heat recovery system in a data centre to reduce energy consumption and emissions. Their results demonstrated significant energy and cost savings as well as improved efficiency, with a projected payback period of 2.5 years. Honda et al. [42] conducted experiments to investigate the effects of mass velocity and condensation temperature difference on local heat transfer during R407C condensation in a horizontal microfine tube to obtain the superficial heat transfer coefficient for the vapour phase, and the combined prediction agreed with the measured values with an error of 9.2%. Rossetto et al. [43] presented a new simple model for predicting heat transfer coefficients in horizontal micro fin tubes during condensation of halogenated and natural refrigerants, validated against a data bank of 3115 experimental heat transfer coefficients. Hargovind et al. [44] used a genetic algorithm to optimize velocity and surface roughness to improve product quality by exploring the effect of pulse on time, wire span, and servo gap voltage on cutting velocity, surface roughness, recast layer, and microhardness of the surface produced. Teng et al. [45] investigated the frictional pressure drop and heat transfer performance of de-ionized water flowing through rectangular microchannels with longitudinal vortex generators (LVGs) results show that heat transfer performance was improved by 12.3–73.8% for microchannels with aspect ratios of 0.0667 and 0.25, respectively, while pressure losses increased by 40.3–158.6% and 6.5–47.7% Hsieh et al. [46] examines the spreading thermal resistance of centrally positioned heat sources and the thermal performance of a flat vapor chamber used for electronic cooling. Parametric studies were conducted, and the results showed a heat removal capacity of 220 W/cm$^2$ with a thermal spreading resistance of 0.2 °C/W for the vapor chamber heat spreader. The study highlights the potential of using flat vapor chambers for efficient electronic cooling. Uzair [47] conducted both experimental performance analysis and deep learning-based modeling to analyze the performance of a closed-loop heat pump dryer that uses R-134a as a secondary fluid and moist sodium

polyacrylate material, also known as Orbeez, as the drying material. The study seeks to understand the behavior of the Orbeez material and its interaction with the heat pump dryer system to improve the efficiency and effectiveness of the drying process

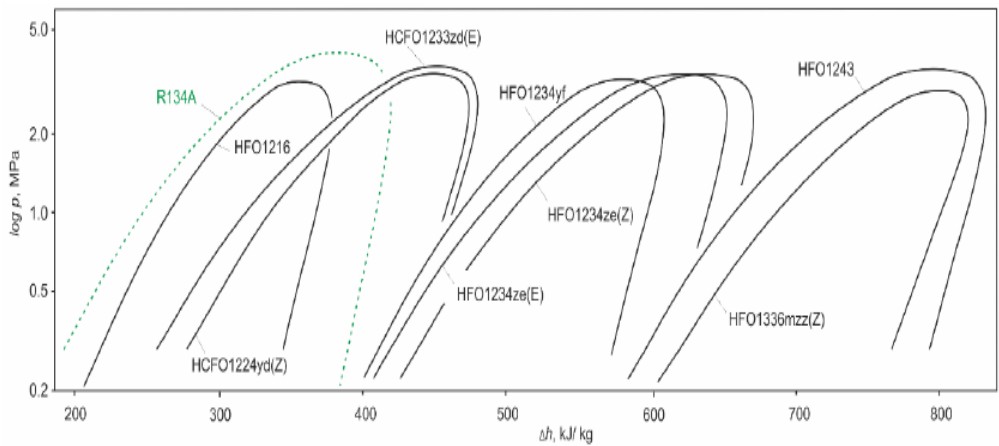

**Figure 1.** Saturation curves of HFO/HCFO refrigerants, collected in p-h diagram [28].

Based on the literature review, refrigerants and blends are used in the simulation investigation as an alternative to R134a and thermodynamic properties and environmental properties are given below in Tables 1–3.

**Table 1.** Thermodynamic and environmental properties of HFC and HFC blends (REFPROP, version 10.0 [23]).

| S. No | Refrigerants/ Properties | Unit | R134a | R404A [R-125/143a/134a] | R407A [R-32/125/134a] | R32 | R152a | R245fa | R227ea | RS50 |
|---|---|---|---|---|---|---|---|---|---|---|
| 1 | Name of the refrigerant | - | 1,1,1,2-Tetrafluoroethane | | | Dichloromethane | 1,1-Difluoroethane | 1,1,1,3,3-Pentafluoropropane | 1,1,1,2,3,3,3-Heptafluoropropano | |
| 2 | Molecular Formula | - | C2H2F4 | C2HF5/C2H3F3/C2H2F4 | CH2F2/C2HF5/C2H2F4 | CH2F2 | C2H4F2 | C3H3F5 | C3HF7 | CH2F2/C2HF5/CH2FCF3/C3HF7/C2H4F2 |
| 3 | Composition (weight share) | - | 100 | 0.44/0.52/0.04 | 20/40/40 | Dichloromethane | 1,1-Difluoroethane | 1,1,1,3,3-Pentafluoropropane | 1,1,1,2,3,3,3-Heptafluoropropano | HFC-32 HFC-125 R134a HFC-227ea HFC-152a |
| 4 | Category (type) | - | HFC | HFC blend | HFC Blend | HFC | HFC | HFC | HFC | HFC Blend |
| 5 | GWP | - | 1430 | 3922 | 2107 | 675 | 124 | 1030 | | 1888 |
| 6 | ODP | - | 0 | 0 | 0 | 0 | 0 | 0 | 0 | 0 |
| 7 | Critical temperature | °C | 101 °C | 72.12 | 82 | 78.1 | 113.26 | 153.86 | 101.75 | 82.4 |
| 8 | Critical pressure | bar | 13.6 | 37.35 | 44.94 | 57.82 | 45.16 | 36.5 | 29.25 | 47.5738 |
| 9 | Normal boiling point | °C | −26.1 °C | −45.74 | −45 | −51.62 | −24.9 | 15.3 | −16 | −46.5 |
| 10 | Molar weight | g/mol | 102.03 | 97.6 | 90.1 | 52.02 | 66.05 | 134.05 | 170.03 | 81.8 |

**Table 2.** Thermodynamic properties of HFC + HFO blends (REFPROP, version 10.0 [23]).

| S. No | Refrigerants/ Properties | Unit | R32/R41/ R1234ze(E) | R134a | R161/R41/ R1234ze(E) | R448A | R449A | R449B | R449C | R450A | R452A | R452B | R454B | R454C | R515A |
|---|---|---|---|---|---|---|---|---|---|---|---|---|---|---|---|
| 1 | Name of the refrigerant | - | | 1,1,1,2-Tetrafluoro-ethane | | HFC32—HFC125—HFC134a—HFO 1234ze—HFO 1234yf | HFC32—HFC125—HFC134a—HFO 1234yf | HFC32—HFC125—HFC134a—HFO 1234yf | HFC32—HFC125—HFC134a—HFO 1234yf | HFC134a—HFO 1234ze (E) | HFC32—HFC125—HFO 1234yf | HFC32—HFC125—HFO 1234yf | HFC32-HFO 1234yf | HFC32-HFO 1234yf | HFCR227ea—HFO 1234ze (E) |
| 2 | Molecular Formula | - | CH2F2/CH3F/C3H2F4 | C2H2F4 | C2H5F/CH3F/C3H2F4 | CH2F2/C2HF5/CH2FCF3/C3H2F4/C3H2F4 | CH2F2/C2HF5/CH2FCF3/C3H2F4 | CH2F2/C2HF5/CH2FCF3/C3H2F4 | CH2F2/C2HF5/CH2FCF3/C3H2F4 | CH2FCF3/C3H2F4 | CH2F2/C2HF5/C3H2F4 | CH2F2/C2HF5/C3H2F4 | CH2F2/C3H2F4 | CH2F2/C3H2F4 | C3HF7/C3H2F4 |
| 3 | Composition (weight share) | - | 0.1/0.9/0 | 100 | 0.8/0.1/0.1 | 26/26/21/7/20 | 24/25/26/25 | 25.2/24.3/23.2/27.3 | 20/20/31/29 | 42/58 | 11/59/30 | 67/7/26 | 68.9/31.1 | 21.5/78.5 | 88/12 |
| 4 | Category (type) | - | HFC + HFO | HFC | HFC + HFO | HFC + HFO | HFC + HFO | HFC + HFO | HFC + HFO | HFC + HFO | HFC + HFO | HFC + HFO | HFC + HFO | HFC + HFO | HFC + HFO |
| 5 | GWP | - | 608 | 1430 | 20 | 1387 | 1397 | 1412 | 1251 | 605 | 2140 | 698 | 466 | 148 | 393 |
| 6 | ODP | - | 0 | 0 | 0 | 0 | 0 | 0 | 0 | 0 | 0 | 0 | 0 | 0 | 0 |
| 7 | Critical temperature | °C | 80.58 | 101 °C | 95.99 | 82.68 | 82.07 | 82.2 | 84.21 | 104.47 | 75.05 | 77.1 | 78.1 | 85.6 | 108.71 |
| 8 | Critical pressure | bar | 58.07 | 13.6 | 53.1 | 45.94 | 44.9 | 45.3 | 43.98 | 38.22 | 40.14 | 52.2 | 52.66 | 43.18 | 35.65 |
| 9 | Normal boiling point | °C | −50.23 | −26.1 °C | −39.77 | −46 | −46 | −46.1 | −44.6 | −23.4 | −47 | −51 | −50 | −46 | −18 |
| 10 | Molar weight | g/mol | 55.02 | 102 | 48.87 | 189.9 | 87.2 | 86.3 | 90.3 | 109.0 | 103.5 | 63.53 | 62.6 | 90.8 | 117.4 |

Table 3. Thermodynamic properties of HFO and PFO and HC (REFPROP, version 10.0 [23]).

| S. No | Refrigerants/ Properties | Unit | R134a | R1216 | R1224yd(Z) | R1233zd(E) | R1234yf | R1234ze(E) | R1234ze(Z) | R1243zf | R1336mzz(Z) | R290 | R600a | RE170 |
|---|---|---|---|---|---|---|---|---|---|---|---|---|---|---|
| 1 | Name of the refrigerant | - | 1,1,1,2-Tetrafluoro-ethane | Hexafluoro-propylene | 1-Chloro-2,3,3,3-tetrafluoro-propene | Trans-1-chloro-3,3,3-Trifluoro-propene | 2,3,3,3-Tetrafluoro-propene | 1,3,3,3-Tetrafluoro-propene | CIS-1, 3,3,3-Tetrafluoro-propene | 3,3,3-Trifluoro-propene | 1,1,1,4,4,4-Hexafluoro-2 butane | Propane | Isobutano | Dimethyl ether |
| 2 | Molecular Formula | - | C2H2F4 | C3F6 | (Z)-CF3-CF=CHCl | C3H2ClF3 | C3H2F4 | C3H2F4 | C3H2F4 | C3ClF3H2 | cis-CF3CH=CHCF3 | CH3CH2CH3 | C4H10 | C2H6O |
| 3 | Composition (weight share) | - | 100 | Hexafluoro-propylene | 1-Chloro-2,3,3,3-tetrafluoro-propene | Trans-1-chloro-3,3,3-Trifluoro-propene | 2,3,3,3-Tetrafluoro-propene | 1,3,3,3-Tetrafluoro-propene | CIS-1, 3,3,3-Tetrafluoro-propene | 3,3,3-Trifluoro-propene | 1,1,1,4,4,4-Hexafluoro-2 butane | Propane | Isobutano | Dimethyl ether |
| 4 | Category (type) | - | HFC | PFO | HFO | HFO | HFO | HFO | HFO | HFO | HFO | HC | HC | HC |
| 5 | GWP | - | 1430 | 17,340 | 4 | 1030 | 1 | 7 | | | 2 | | | |
| 6 | ODP | - | 0 | 0 | 0 | 0 | 0 | 0 | 0 | 0 | 0 | 0 | 0 | 0 |
| 7 | Critical temperature | °C | 101 | 85.8 | 155.54 | 166.45 | 94.3 | 109.36 | 150.2 | 103.7 | 171.35 | 96.7 | 134.7 | 127.2 |
| 8 | Critical pressure | bar | 13.6 | 31.49 | 33.37 | 36.23 | 33.82 | 35.34 | 35.3 | 35.17 | 29.03 | 42.5 | 36.3 | 53.37 |
| 9 | Normal boiling point | °C | −26.1 | −29.6 | 15 | 18.31 | −29.48 | −19 | 9.8 | −25.42 | 33.4 | −42 | −12 | −24.78 |
| 10 | Molar weight | g/mol | 102.03 | 150.03 | 148.5 | 130.5 | 114 | 114 | 114.04 | 96.05 | 164 | 44.1 | 58.12 | 46.07 |

## 2. Experimental Apparatus and Test Conditions

Compressor, condenser, evaporator, and expansion valves are the main components of any simple vapour compression refrigeration system to sustain the cooling load, whereas some applications require different temperatures for different sections. Tests have been carried out in a controlled environment with an environment temperature of 18 °C and evaporator and condenser air-flow discharge conditions. The pressure of refrigerant in the condenser and the evaporator, the temperatures in the refrigeration loop, and the compressor power consumption data for each of the tests were recorded with a period of 10 s per measurement in the dynamic cooling process from 15 °C to −10 °C as measured as the outlet of the evaporator. The experiment started with R134a to set up the base reference, the thermodynamic properties of the refrigerants were obtained from the NIST thermodynamic properties of refrigerants and refrigerant mixtures database [23]. The vapour compression refrigeration cycle is based on the following factors:

- Refrigerant flow rate.
- Type of refrigerant used.
- Kind of application viz air-conditioning, refrigeration, dehumidification, etc.
- The operation design parameters.
- The system equipment/components proposed to be used in the system.

A single-stage vapour compression system was used to generate data to verify the model. The system was equipped with a variable speed reciprocating compressor, variably sized evaporator and condenser, manually adjusted throttling valve, and a liquid-line/suction-line heat exchanger, which could be included or bypassed The evaporator and condenser were of the annular design arranged in the counter-current configuration; the refrigerant flowed in the enhanced inner tube (copper), while the HTF flowed in the smooth annular space. The heat exchangers' size could be adjusted by changing the number of active refrigerant tubes; this feature enabled heat flux control. The apparatus was set to achieve evaporation and condensation saturation temperatures nominal to air-source heat pumps and the HTF inlet and outlet temperatures were used to obtain these evaporation and condensation temperatures using R134a and a mid-range compressor speed, 1800 rev·min$^{-1}$. Four additional data sets at each rating test (total of 12) were generated by holding the HTF inlet temperature constant as the system capacity was varied via compressor speed, (1400 to 2200) rev·min$^{-1}$; readings are mentioned in observation Table 4, and care was taken to configure other evaporator and condenser operating conditions (beyond refrigerant saturation temperature) to closely resemble those of a typical air-to-air heat pump. Specifically, the heat fluxes were within (5 to 9) kW·m$^{-2}$ and (5 to 10) kW·m$^{-2}$ for the evaporator and the condenser, respectively. Additionally, the ratios of HTF thermal resistance to total heat exchanger thermal resistance were nominally 0.8 and 0.6 for the evaporator and condenser, respectively; these values are representative of air-to-air heat pumps where the air side (i.e., HTF side) thermal resistance dominates. The thermal resistance ratios were enforced by the selection of HTF mass flow rates; the HTF mass flow rates were held constant for all tests at 0.098 kg·s$^{-1}$ for the condenser and 0.131 kg·s$^{-1}$ for the evaporator. The subcooling and superheat were controlled to (2 to 3) K and (3 to 6) K, respectively. More details about these tests, including the uncertainty calculation (95% confidence level) for the COP (0.35%), capacity (0.2%), and $Q_{vol}$ (1.5%).

**Table 4.** Observation table for R134a as refrigerant.

| S. No | Energy Meter Reading for 10 Rev in Sec. | Compressor Inlet Pressure, $P_1$ (Bar) | Compressor Outlet Pressure, $P_2$ (Bar) | Refrigerant Temperature at Inlet of Compressor, $T_1$ (°C) | Refrigerant Temperature at Outlet of Compressor, $T_2$ (°C) | Refrigerant Temperature at Inlet of Expansion Valve, $T_3$ (°C) | Refrigerant Temperature at Outlet of Expansion Valve, $T_4$ (°C) | Water Temperature in Evaporator, $T_5$ (°C) |
|---|---|---|---|---|---|---|---|---|
| 1. | 8.5 | 6.4 | 8.3 | 15.2 | 90.3 | 49.3 | 6.3 | 8.6 |
| 2. | 9.7 | 6.3 | 8.5 | 16.5 | 91.6 | 51.1 | 5.9 | 7.9 |
| 3. | 10.6 | 6.5 | 8.7 | 17.8 | 92.7 | 52.3 | 5.2 | 7.2 |

From the above observation table, the calculation is performed to find out the refrigerating effect and work completed by the compressor

**Calculation-**

$$\begin{aligned}
\text{Work done by compressor (WD)} &= \frac{\text{No. of revolutions in energy meter} * 3600}{\text{Time taken in energy meter} * \text{Emc}} \\
&= \frac{10 * 3600}{85 * 750} \\
&= 0.5647 \text{ KW}
\end{aligned} \tag{1}$$

In Case of mass of water in the chiller,

$$\begin{aligned}
\text{Refrigerating effect(RE)} &= \frac{m_w * c_p * \Delta T}{\text{Time taken for drop in initial and final temperature}} \\
&= \frac{11 * 4.187 * 8.6}{10 * 60} \\
&= 0.660 \text{ KW}
\end{aligned} \tag{2}$$

$$\begin{aligned}
\text{Actual Coefficient of performance (COP}_{\text{actual}}) &= \frac{RE}{WD} \\
&= \frac{0.660}{0.5647} \\
&= 1.169
\end{aligned} \tag{3}$$

For the variations, the pressure and the temperature ranges are changed,

$P_1$ = 6.4 Bar
$P_2$ = 8.3 Bar
$T_1$ = 24.3 °C, $h_1$ = 373.13 KJ/Kg
$T_2$ = 34.7 °C, $h_2$ = 425.84 KJ/Kg
$T_3$ = 31.7 °C, $h_3$ = 243.9 KJ/Kg
$T_4$ = 15.3 °C, $h_4$ = 220 KJ/Kg

$$\begin{aligned}
\text{Theoretical coefficient of performance} \left(\text{COP}_{\text{theory}}\right) &= \frac{h_1 - h_4}{h_2 - h_1} \\
&= \frac{373.13 - 220}{425.84 - 373.13} \\
&= 2.90
\end{aligned} \tag{4}$$

$$\text{Exergy at any point can be calculated as}: \dot{ex} = \dot{m}[(h - h_{\text{air}}) - T_{\text{air}}(s - s_{\text{air}})] \tag{5}$$

Accordingly, the total exergy destruction is sum of the exergy destruction in each of components and is written as:

$$\dot{Exd}_{tot} = \dot{Exd}_{eva} + \dot{Exd}_{con} + \dot{Exd}_{com,HTC} + \dot{Exd}_{com.LTC} + \dot{Exd}_{exp,HTC} + \dot{Exd}_{exp,LTC} \tag{6}$$

*2.1. Description of CYCLE_D-HX 2.0 Model*

The CYCLE_D-HX 2.0 model is a simulation tool that is used to analyse the performance of vapour compression cycles. It is based on the concept of using temperature profiles of the heat sink and heat source, and $\Delta T_{\text{hx}}$ for the evaporator and condenser. This approach enables the model to account for refrigerant thermophysical properties, pressure drop, and heat transfer coefficient on the cycle performance on a relative basis. The simulated system consists of a compressor, condenser, adiabatic expansion device, and evaporator. The compressor is represented by the isentropic efficiency, volumetric efficiency, and electric motor efficiency. The evaporator and condenser can be either counter-flow, crossflow, or parallel-flow, and are represented by their $\Delta T_{\text{hx}}$. The solution sequence starts with estimated values of saturation temperatures in the evaporator and condenser. Based on the established thermodynamic cycle with refrigerant temperature profiles and HTF (heat transfer fluid) temperature profiles, the model calculates $\Delta T_{\text{hx}}$ and compares them to the values specified as input. The model iterates evaporator and condenser saturation temperatures until it achieves the specified $\Delta T_{\text{hx}}$ values within a convergence parameter. The CYCLE_D-HX 2.0 model is a comprehensive tool that can be used to analyse the performance of different types of vapour compression cycles. It includes enhanced cycle options such as a liquid-line/suction-line heat exchanger. The model has been extensively

tested and validated against experimental data, and its accuracy and reliability have been demonstrated in several research publications.

For each iteration step of saturation temperatures, CYCLE_D-HX 2.0 calculates heat exchangers' $\Delta T_{hx}$ using Equation (7)

$$\frac{1}{\Delta T_{hx}} = \frac{Q_1}{Q_{hx} \Delta T_1} + \frac{Q_2}{Q_{hx} \Delta T_2} = \frac{1}{Q_{hx}} \sum \frac{Q_i}{\Delta T_i} \tag{7}$$

The equation presented in the statement calculates $\Delta T_{hx}$ as a harmonic mean weighted with the fraction of heat transferred in individual sections of the heat exchanger, assuming a constant overall heat transfer coefficient throughout the heat exchanger. This approach enables the model to account for the variations in heat transfer rate along the length of the heat exchanger. At the beginning of each iteration, the model calculates $\Delta T_{hx}$ based on the sections corresponding to the subcooled liquid, two-phase, and superheated regions. The model repeatedly bisects each subsection until the $\Delta T_{hx}$ obtained from two consecutive evaluations agree within a convergence parameter. This iterative process ensures that the model achieves a high degree of accuracy in calculating the performance of the vapour compression cycle. Alternatively, the heat exchangers can be characterised by the overall heat conductance $UA_{hx}$, which is a measure of the heat transfer rate per unit temperature difference across the heat exchanger. If this input option is used, the model calculates the specified $\Delta T_{hx}$ from the basic heat transfer relation, which relates the heat transfer rate to the temperature difference and the overall heat transfer coefficient. This approach provides an alternative method to specify the heat exchanger performance, which may be more convenient in some cases. This input option is used, the model calculates the specified $\Delta T_{hx}$ from the basic heat transfer relation. If this input option is used, the model calculates the specified $\Delta T_{hx}$ from the basic heat transfer relation,

$$\Delta T_{hx} = \frac{Q_{hx}}{UA_{hx}} \tag{8}$$

where $Q_{hx}$ is the product of refrigerant mass flow rate and enthalpy change in the evaporator or condenser, as appropriate. Representation of heat exchangers by their $UA_{hx}$ allows for the inclusion of heat transfer and pressure drop characteristics in comparative evaluations of different refrigerants. For this purpose, CYCLE_D-HX 2.0 considers that the total resistance to heat transfer in a heat exchanger, $R_{hx}$, consists of the resistance on the refrigerant side $R_r$, and combined resistances of the heat exchanger material and HTF [$R_{tube} + R_{HTF}$]:

$$R_{hx} = \frac{1}{UA_{hx}} = R_r + [R_{tube} + R_{HTF}] \tag{9}$$

$$\text{where, } R_r = \frac{1}{(h_r * A_{hx})} \tag{10}$$

The resistances [$R_{tube} + R_{HTF}$] are independent of the refrigerant and are assumed to be independent of operating conditions. Their combined value can be calculated from $UA_{hx}$ and hr values using performance measurements obtained in a laboratory on a system of interest CYCLE_D-HX 2.0calculates [$R_{tube} + R_{HTF}$] within its "reference run" and stores its value for use in subsequent simulation runs for the calculation of $UA_{hx}$ characterising the heat exchanger with a new refrigerant or operating conditions. CYCLE_D-HX 2.0 requires the following operational input data for the "reference run": Figure 2. Then, several inputs to simulate the performance of the vapour compression cycle. These inputs include the HTF inlet and outlet temperatures for the evaporator and condenser, $\Delta T_{hx}$ for the evaporator and condenser to achieve the desired measured evaporator and condenser saturation temperatures, evaporator superheat and pressure drop, and condenser subcooling and pressure drop. The "reference run" inputs include compressor isentropic and volumetric efficiencies and electric motor efficiency. The isentropic efficiency of the compressor can be dependent on the compression ratio, and CYCLE_D-HX 2.0 offers the option of accounting for this

dependence. When screening different refrigerants, the model uses a set of thermophysical properties and correlations to simulate their behaviour in the vapour compression cycle. The accuracy and reliability of the model's predictions depend on the quality of the input data and the assumptions made in the simulation. Therefore, it is important to validate the model against experimental data and adjust the inputs and assumptions accordingly. Equation (11) takes into account the change in isentropic efficiency with the pressure ratio in a consistent way.

$$\eta_s = C - 0.05\theta \tag{11}$$

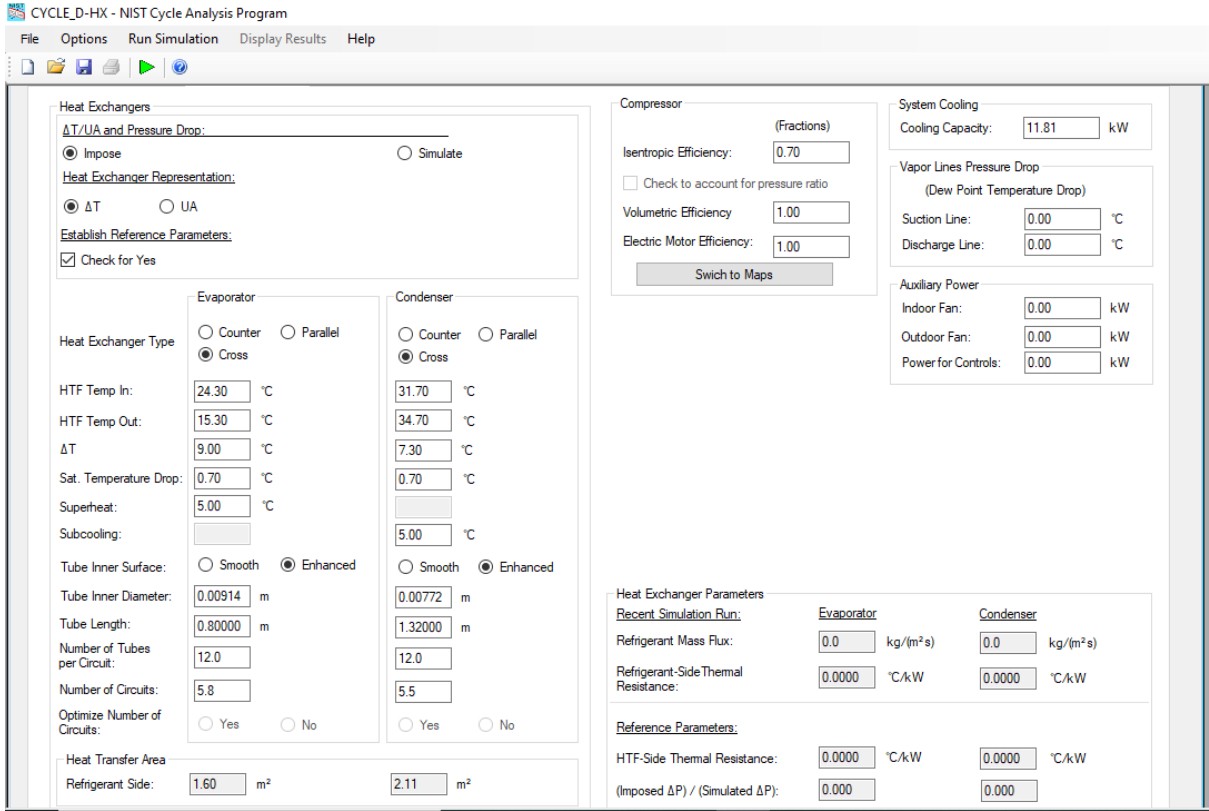

**Figure 2.** Input parameters (CYCLE_D_HX 2.0).

C is a constant calculated within the "reference run" using the isentropic efficiency.

### 2.2. CYCLE_D-HX 2.0 Simulation Model Validation

We used the data from the cooling of the R134a test to carry out the CYCLE_D-HX 2.0 "reference run". The "reference run" inputs included the 11.81 kW capacity, the evaporator $\Delta T_{hx} = 9\,^\circ C$, the condenser $\Delta T_{hx} = 7.3\,^\circ C$, and pressure drops of 33 kPa and 45 kPa for the condenser and evaporator, respectively. We then executed simulations of the remaining Cooling A, Cooling B, and Heating H1 rating tests. The capacities, compressor isentropic and volumetric efficiencies, superheat and subcooling, discharge and suction line pressure drops, and HTF inlet and outlet temperatures were input based on measurements from each test. We evaluated the percentage deviation between the simulation and the experimental results using Equation (12).

$$E = \frac{\Pi Experimental - \Pi Simulation}{\Pi Experimental} \cdot 100\% \tag{12}$$

where $\Pi$ is any parameter of interest.

The deviations for COP, $Q_{vol}$, $p_{evap}$, and $p_{cond}$. Most of the deviations are within 4%. The largest deviation (7.4%) is for the Cooling B test at the highest (2200 rev·min$^{-1}$)

compressor speed; this operating condition yielded about a 20% increase in refrigerant mass flow rate and capacity over the "reference run".

*2.3. Exergy Analysis*

Exergy analysis is a powerful tool used to evaluate the performance of a refrigeration system. It allows us to identify the sources of irreversibility and inefficiencies in the system and helps us to determine where improvements can be made. The vapour compression refrigeration system (VCRS) is a common refrigeration system used in many applications. It consists of a compressor, condenser, expansion valve, and evaporator. The refrigerant circulates through these components and undergoes phase changes as it absorbs and releases heat, resulting in the cooling of the desired space or product. To conduct an exergy analysis of a VCRS, we can use the following steps-

1.  Define the system boundary and identify the components within the boundary. This would typically include the compressor, condenser, expansion valve, and evaporator.
2.  Calculate the thermodynamic properties of the refrigerant at various points in the system, such as the temperature, pressure, and specific enthalpy. This can be completed using thermodynamic tables or software.
3.  Calculate the exergy at each component and at each state point using the following equation-

$$\text{Exergy} = (\text{enthalpy} - \text{enthalpy}_{\text{ref}}) - T_{\text{ref}}(\text{entropy} - \text{entropy}_{\text{ref}}) \tag{13}$$

where enthalpy and entropy are the thermodynamic properties of the refrigerant, enthalpy$_{\text{ref}}$ and entropy$_{\text{ref}}$ are the thermodynamic properties of the refrigerant at a reference state (typically the dead state or environment), and $T_{\text{ref}}$ is the reference temperature.

1.  Calculate the exergy destruction at each component by taking the difference between the exergy input and output. This represents the amount of exergy lost due to irreversibilities and inefficiencies in the component.
2.  Calculate the overall system exergy efficiency, which is the ratio of the exergy output to the exergy input. This represents the percentage of the available exergy that is being used to perform useful work.

By conducting an exergy analysis of a VCRS, we can identify the components and processes that are contributing the most to exergy destruction and inefficiencies. This can help us to make improvements to the system design, such as using more efficient components, optimising operating conditions or implementing waste heat recovery systems to reduce the amount of exergy lost to the environment.

## 3. Results and Discussion

The above calculations are validated with simulation software CYCLE_D-HX 2.0-NIST cycle analysis program for investigation over different refrigerants the input parameters are taken, and three groups of refrigerants are made for result analysis of variations in different parameters that can affect the system.

Scheme 1 shows pressure variation for 31 refrigerants at the compressor shell inlet. A blended refrigerant mixture of R32/R41/R1234ze(E) has a maximum pressure of 1448.6 KPa, and R1336mzz(Z) has a minimum pressure of 38.7 KPa. Inlet pressure affects refrigerating effect and efficiency, Low inlet pressure reduces refrigerant density and power consumption and is desirable for vapour compression refrigeration systems. Lower pressure can be achieved by installing fouled inlet filters or changing barometric pressure. Reduced weight flow at the inlet decreases power consumption or horsepower. Pressure at the compressor shell inlet for R134a is recorded as 409.5 KPa. The highest point exergy of 559.33 J is recorded by R290.

**Scheme 1.** Comparison of other refrigerants on compressor shell inlet pressure with R134a.

Figure 3 displays the results of the enthalpy and pressure variation for different refrigerants. The maximum pressure and enthalpy are obtained for R-32, which has a pressure of 1095.1 KPa and an enthalpy of 523.2 kJ/kg. In addition, the HFC blended refrigerant and HFO blended refrigerant R32/R41/R1234ze(E) show the maximum pressure and enthalpy of 1448.6 KPa and 481.3 kJ/kg, respectively. Furthermore, the HC refrigerant R290 exhibits a maximum pressure and enthalpy of 629.9 KPa and 594.4 kJ/kg at the compressor shell inlet temperature of 15 °C.

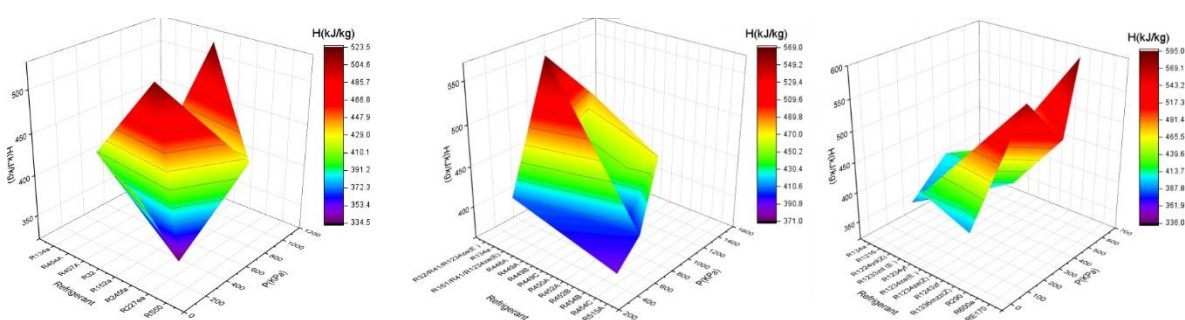

**Figure 3.** Comparison of other refrigerants on compressor shell inlet pressure vs. enthalpy with R134a.

Scheme 2 illustrates the variation of work completed by the compressor for different refrigerants and refrigerant mixtures in the system. The investigation revealed that R1216

and R227ea refrigerants require the least amount of work in the compressor, at 17.64 kJ/kg and 17.37 kJ/kg, respectively. On the other hand, refrigerant RE170 demands a high amount of work at 63.43 kJ/kg in the compressor. Compressors play a crucial role in the system as they receive low-pressure refrigerant from the evaporator and compress it into the high-pressure refrigerant. The efficiency of the system is highly dependent on the work completed by the compressor, and the lower work completed by the compressor indicates a higher level of efficiency.

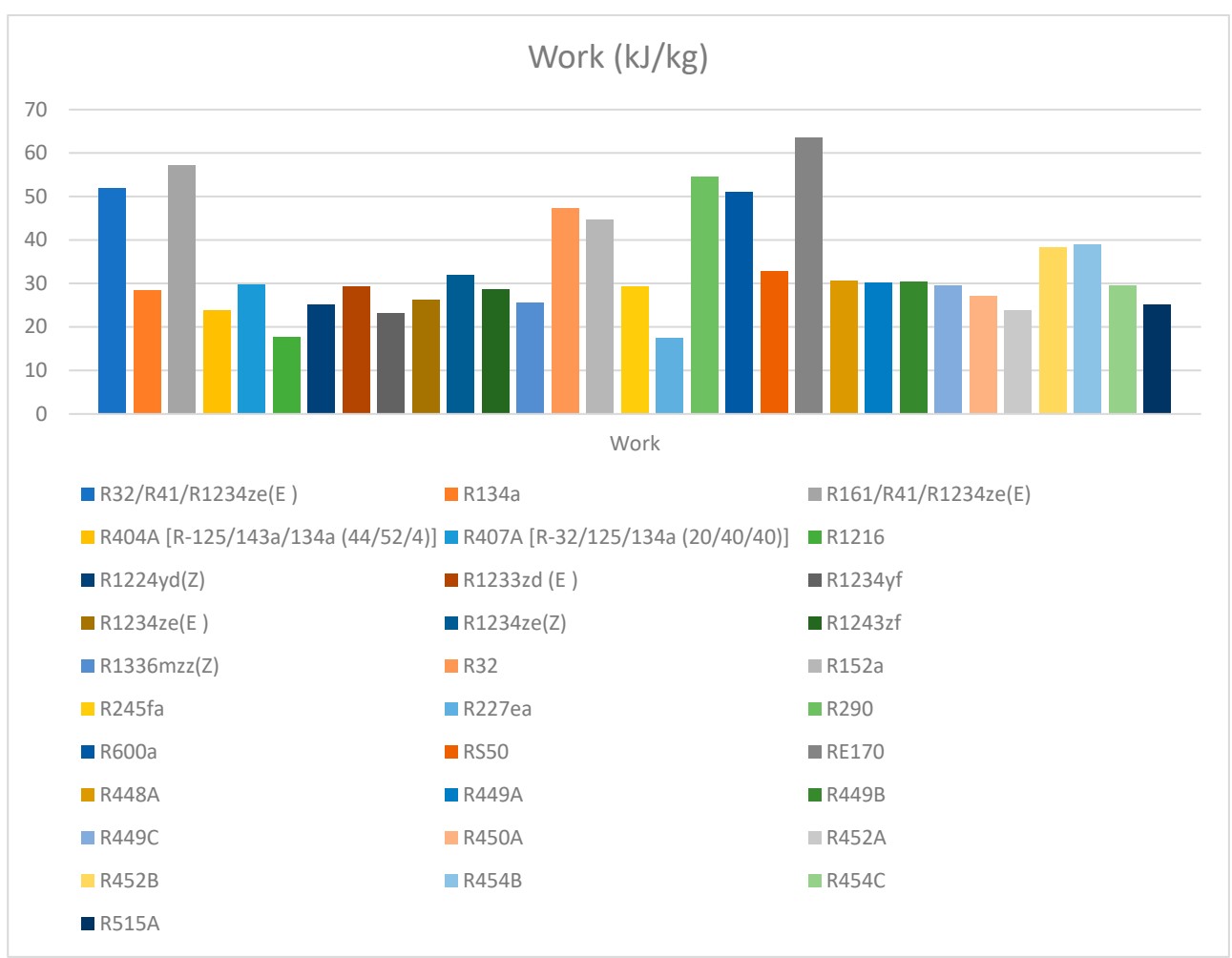

**Scheme 2.** Comparison between eco-friendly refrigerants with R134a.

Figure 4 displays the relationship between the work completed by the compressor and heat supplied in the evaporator for different refrigerants at the same input conditions. The results show that R-32, HFO blended refrigerant R161/R141/R1234ze(E), and HC refrigerant RE170 have the maximum heat supplied in the evaporator and work completed values of 259.3 kJ/kg and 47.34 kJ/kg, 299.43 kJ/kg and 57.22 kJ/kg, and 368.75 kJ/kg and 63.43 kJ/kg, respectively, at the same temperature of 15°C in the evaporator unit. This information is critical in selecting the most efficient refrigerant for a particular refrigeration system.

Scheme 3 depicts the heat transfer variation in the evaporator unit for different refrigerants and refrigerant mixtures. The investigation revealed that the maximum heat transfer was achieved by RE170 at 368.75 kJ/kg, which is 23% higher than that of R134a. On the other hand, refrigerants R1216 and R227ea recorded the lowest heat transfer at 93 and 93.55 kJ/kg, respectively. A higher value of heat transfer in the evaporator is desirable as it indicates faster heat transfer. Highest point exergy of 279.31 J is recorded by R290.

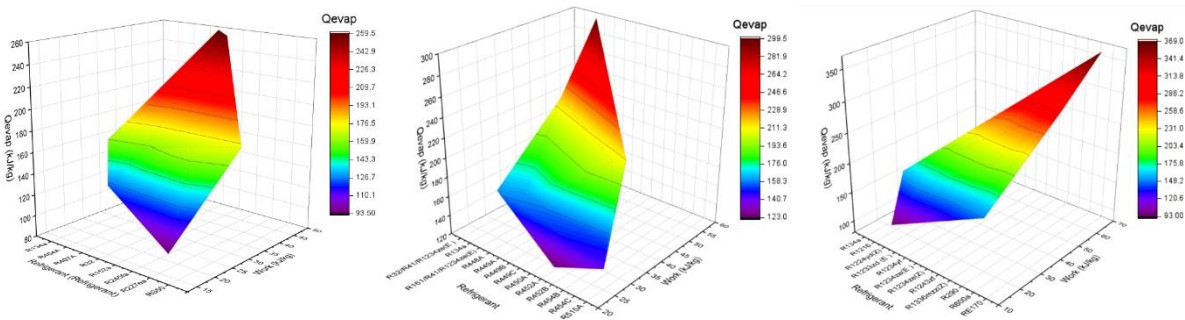

**Figure 4.** Comparison of refrigerants between work completed and Q_{evap} on with R134a.

### Qevap kJ/kg

Qevap

- R32/R41/R1234ze(E )
- R134a
- R161/R41/R1234ze(E)
- R404A [R-125/143a/134a (44/52/4)]
- R407A [R-32/125/134a (20/40/40)]
- R1216
- R1224yd(Z)
- R1233zd (E )
- R1234yf
- R1234ze(E )
- R1234ze(Z)
- R1243zf
- R1336mzz(Z)
- R32
- R152a
- R245fa
- R227ea
- R290
- R600a
- RS50
- RE170
- R448A
- R449A
- R449B
- R449C
- R450A
- R452A
- R452B
- R454B
- R454C
- R515A

**Scheme 3.** Comparison between eco-friendly refrigerants with R134a.

Figure 5 illustrates the relationship between specific volume and pressure in the evaporator unit for different refrigerants and refrigerant mixtures. The results show that R32 has the highest pressure and enthalpy, with maximum values of 1118.1 KPa and 263.8 kJ/kg, respectively. The blended refrigerant mixture of R32/R41/R1234ze(E) recorded the highest pressure of 1479.8 KPa and the highest enthalpy of 269 kJ/kg, while R161/R41/R1234ze(E) recorded the maximum enthalpy of 269 kJ/kg. Refrigerants R290 and R134a had the highest pressure and enthalpy of 642.5 KPa and 249.1 kJ/kg, respectively. Overall, the graph

provides insights into the specific volume–pressure relationship of different refrigerants and their mixtures in the evaporator unit.

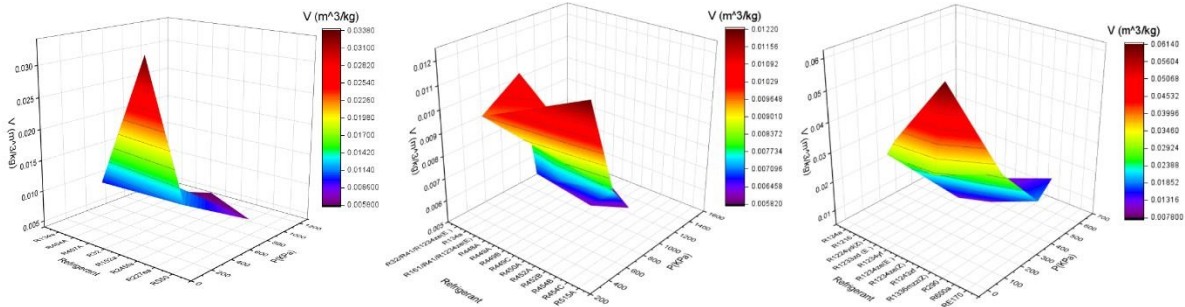

**Figure 5.** Comparison of refrigerants between inlet pressure vs. specific volume with R134a.

Scheme 4 displays the heat transfer rate variation for different refrigerants in the condenser unit. A high heat transfer rate is desirable to convert high-pressure vapour refrigerant into a high-pressure liquid. The highest rate is recorded for RE170 at 432.18 KJ/Kg, which is 23.5% higher than R134a. A good refrigerant should have a high heat transfer rate during condensation for efficient cooling. The highest point exergy of 511.72 J is recorded by R290.

$Q_{condensar}$ kJ/kg

Legend:
- R32/R41/R1234ze(E )
- R134a
- R161/R41/R1234ze(E)
- R404A [R-125/143a/134a (44/52/4)]
- R407A [R-32/125/134a (20/40/40)]
- R1216
- R1224yd(Z)
- R1233zd (E )
- R1234yf
- R1234ze(E )
- R1234ze(Z)
- R1243zf
- R1336mzz(Z)
- R32
- R152a
- R245fa
- R227ea
- R290
- R600a
- RS50
- RE170
- R448A
- R449A
- R449B
- R449C
- R450A
- R452A
- R452B
- R454B
- R454C
- R515A

**Scheme 4.** Comparison between eco-friendly refrigerants with R134a.

Figure 6 presents the results of the variation in enthalpy to pressure in the condenser unit for different refrigerants and refrigerant mixtures at the same input conditions. The graph shows that the maximum pressure of 1882.7 KPa is recorded for HFC blend RS50, while the highest enthalpy of 531.6 kJ/kg is recorded for HFC refrigerant R152a. The blended HFC and HFO refrigerant of R452b recorded a maximum pressure of 2309.1 KPa, and a maximum enthalpy of 576.3 kJ/kg is recorded for R161/R41/R1234ze(E). Furthermore, refrigerant R290 recorded the highest pressure and enthalpy of 1394 KPa and 614.8 kJ/kg, respectively.

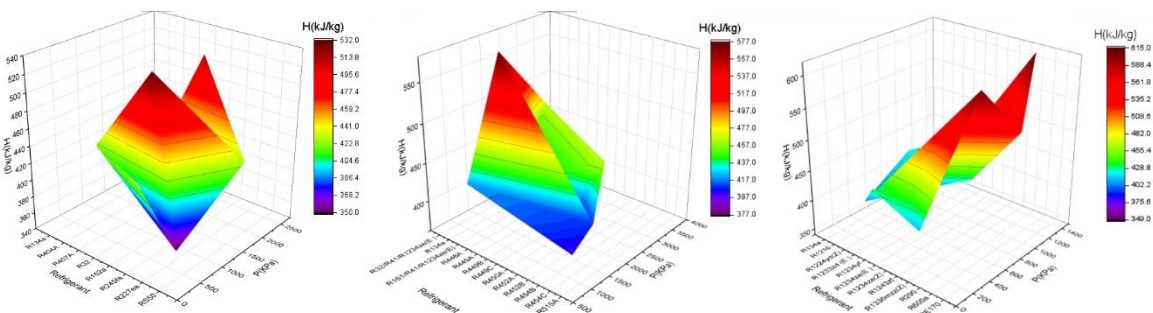

**Figure 6.** Comparison of refrigerants in condenser between pressure vs. enthalpy with R134a.

Scheme 5 displays the cooling rate of various refrigerants and refrigerant blends in the system. A higher cooling rate is desirable for a specific volume, and the investigation found that a mixture of refrigerants R32/R41/R1234ze(E) provides the highest cooling rate of 8290.5 kJ/m$^3$, which is 27% higher than the refrigerant R134a. The refrigerant R1336mzz(z) recorded the lowest cooling rate at 403.4 kJ/m$^3$.

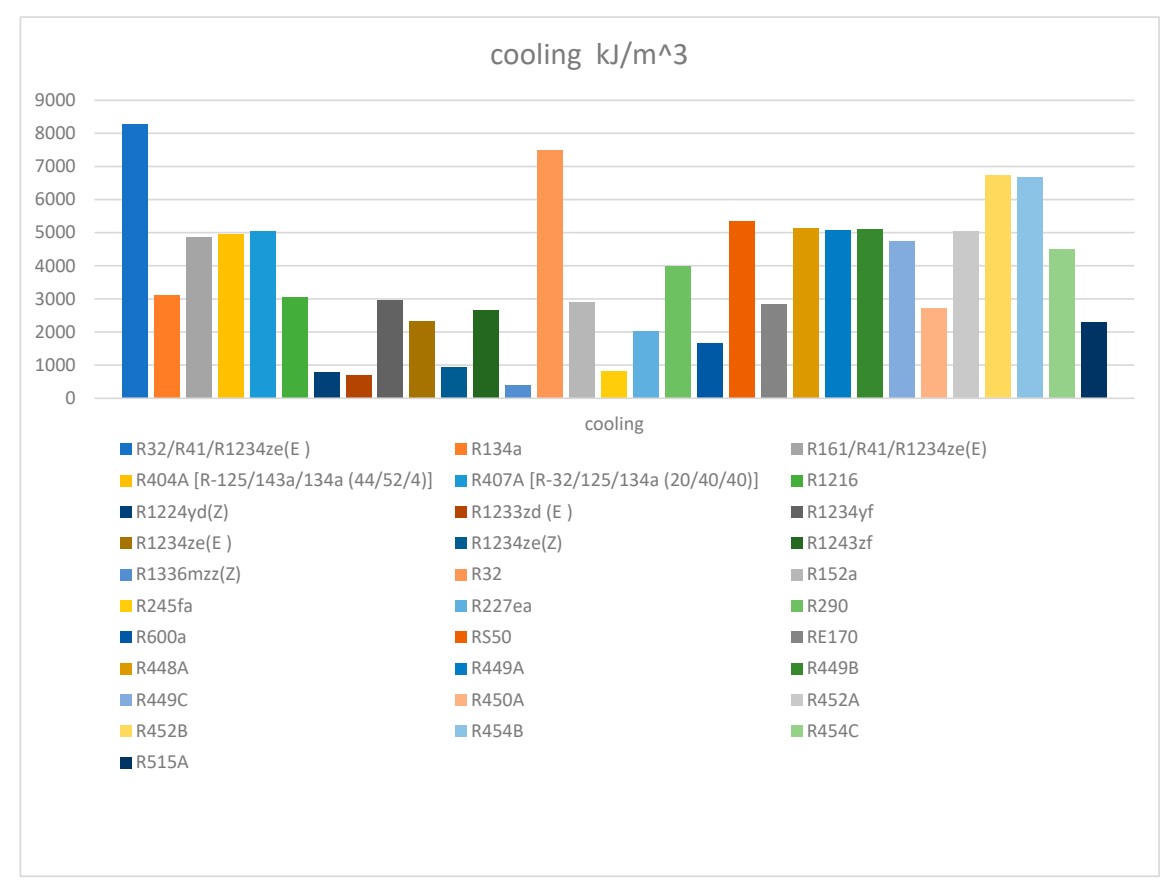

**Scheme 5.** Comparison of the cooling capacity of other refrigerants with R134a.

Figure 7 illustrates the variations in cooling with the same coefficient of performance (COPc) for different refrigerants and blends. The results of the investigation show that the highest rate of cooling is achieved by HFC refrigerant R32 at 7478.5 kJ/m$^3$, while the lowest rate of cooling is recorded by HFC refrigerant R245fa at 802 kJ/m$^3$. For blended refrigerants of HFC and HFO, the highest rate of cooling is recorded by R32/R41/R1234ze(E) at 8290.5 kJ/m$^3$, and the lowest is recorded by R515A at 2303 kJ/m$^3$. In the third category of refrigerants from PFO, HC, and HFO, the highest rate of cooling is recorded by R290 at 4000.8 kJ/m$^3$, and the lowest is recorded by R1336mzz(z) at 403 kJ/m$^3$.

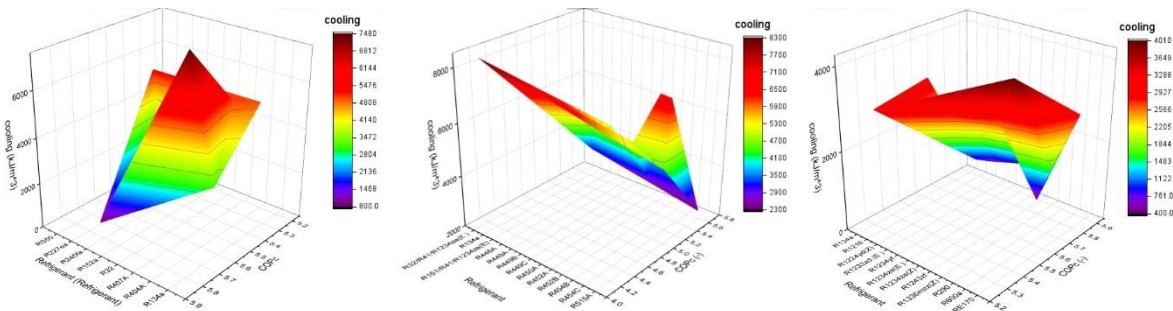

**Figure 7.** Comparison of refrigerants between cooling effect vs. COPc with R134a.

Scheme 6 shows the coefficient of performance (COP$_c$) of different pure refrigerants and blended refrigerants in the cooling system. It is observed that pure refrigerants such as RE170, R245fa, R1234ze, R1233zd(E), and R1224yd(Z) have a higher efficiency of 5.8, which is slightly higher than the efficiency of R134a. The lowest coefficient of performance (COP$_c$) is recorded as 4.178 for the blend of R32/R41/R1234ze(E). The other selected refrigerants and blends have almost the same performance, and they can be easily used as an alternative to R134a.

Scheme 7 depicts the variation in volume flow rate during the compression process for different refrigerants and blended refrigerants. R1336mzz(Z) is found to have the highest volume flow rate of 105.3 m$^3$/h, which is 80% higher than R134a. On the other hand, the blended refrigerant R32/R41/R1234ze(E) has the least value of 5.128 m$^3$/h in the graph. The volume flow rate affects the speed of the compressor, the amount of refrigerant, and the refrigerant flow through the evaporator. A thorough investigation revealed that higher flow rates lead to a better distribution of refrigerant in the system. As the refrigerant charge increases, the temperature decreases in the compressor. Consequently, the load on the compressor decreases and the discharge temperature of the refrigerant increases.

Figure 8 represents the variation in compressor suction volume flow rate for different refrigerants and blends at the same compressor power of 2 kW. The graph shows that HFC refrigerant R245fa has the highest compressor suction volume flow rate of 53.013 m$^3$/h while HFC refrigerant R32 has the lowest compressor suction volume flow rate of 5.685 m$^3$/h. For blended refrigerants of HFC and HFO, HFC refrigerant R515a has the highest compressor suction volume flow rate of 18.5 m$^3$/h while the blend of HFC and HFO refrigerants R32/R41/R1234ze(E) has the lowest compressor suction volume flow rate of 5.1 m$^3$/h. Among HFO refrigerants, R1336mzz(Z) has the highest compressor suction volume flow rate of 105.39 m$^3$/h and for HC refrigerant, R290 has the lowest compressor suction volume flow rate of 10.6 m$^3$/h. Compressor suction volume flow rate affects the refrigerant flow through the evaporator and higher flow rates result in better distribution of refrigerant in the system.

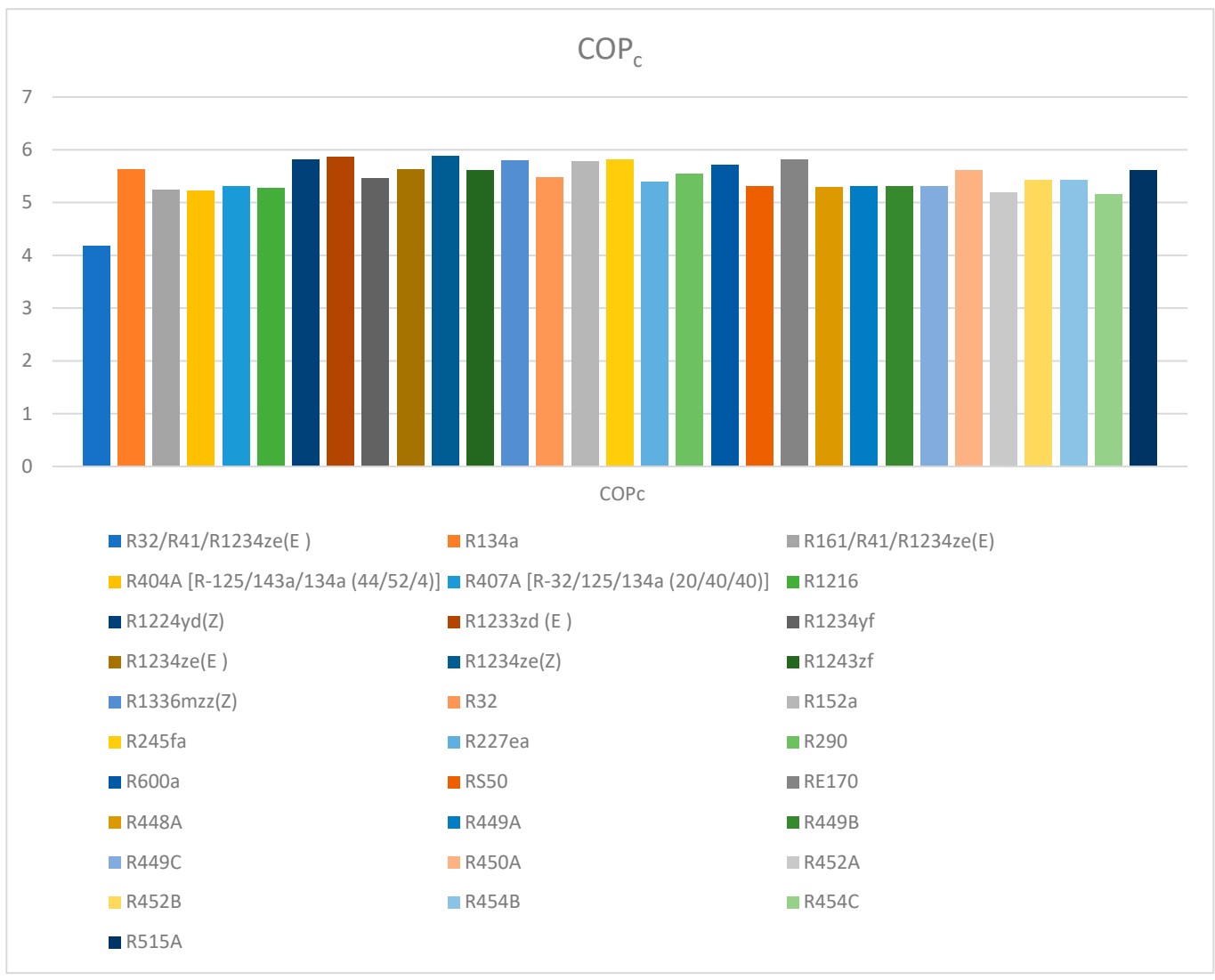

**Scheme 6.** Comparison between eco-friendly refrigerants with R134a.

## Compr. suc. vol. flow rate(m^3/h)

**Scheme 7.** Comparison between eco-friendly refrigerants with R134a.

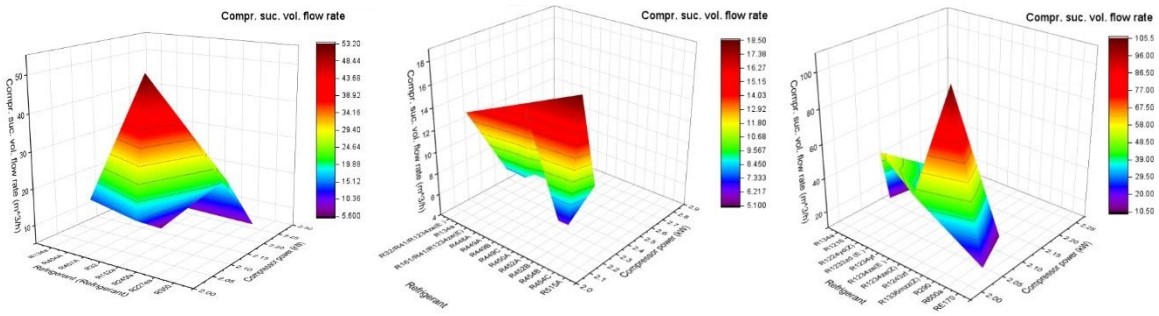

**Figure 8.** Comparison of refrigerants between power vs. volume flow rate with R134a.

### 4. Conclusions

On the basis of experimental and simulation investigations, the following conclusions are made

- Maximum pressure at the compressor is recorded by the blended refrigerant mixture of R32/R41/R1234ze(E) at 1448.6 kPa and the minimum value is recorded by R1336mzz(Z) at 38.7 KPa.
- Refrigerants R1216 and R227ea consume minimum work of 17.64 kJ/kg and 17.37 kJ/kg, respectively, and refrigerant RE170 requires a high amount of work of 63.43 kJ/kg in the compressor.
- The maximum rate of heat transfer in the evaporator is recorded by RE170 as 368.75 kJ/kg which is 23% higher than R134a and the least value is recorded by refrigerants R1216 and R227ea at 93 and 93.55 kJ/kg, respectively.
- The highest rate of heat transfer in the condenser unit is obtained by RE170 at 432.18 kJ/kg which is around 23.5% higher than R134a.
- R32/R41/R1234ze(E) recorded the highest rate of cooling of 8290.5 kJ/m3 recorded and it is 27% higher than the refrigerant R134a.
- Pure refrigerants RE170, R245fa, R1234ze, R1233zd(E), and R1224yd(Z) have higher efficiency of 5.8 which is slightly higher than the efficiency of R134a.
- A higher compressor suction volume flow rate is attained by R1336mzz(Z) of 105.3 $m^3$/h it is 80% higher than the R134a and blended refrigerant R32/R41/R1234ze(E) is noted as the least value of 5.128 $m^3$/h.

These findings highlight the significant differences between the various refrigerants in terms of their thermodynamic properties and performance. These results have important implications for the selection of refrigerants in various cooling applications, and the study provides valuable insights for researchers and practitioners in the field of refrigeration and air conditioning. Furthermore, the manuscript highlights the importance of the work in addressing the environmental concerns associated with traditional refrigerants such as R134a. As the global focus on reducing greenhouse gas emissions intensifies, the use of new and more environmentally friendly refrigerants is becoming increasingly important. The study provides an important contribution to this effort, and its findings can be used to inform the development of policies and regulations aimed at promoting the adoption of environmentally friendly refrigerants in industrial refrigeration applications.

## 5. Problems in the System due to Blending or Mixing

During the mixing of two or more refrigerants, each other following problems occur-

- During the running condition of the system, the effectiveness of the system is reduced due to the phase change in refrigerant in the condenser and evaporator unit as the properties of the blended refrigerant change.
- Due to uncertain non-isothermal behaviour and a mixture of refrigerants, the manufacturers are unable to design and select the appropriate component for the system from the catalogue.
- Only specific heat exchangers such as a flat plate and counter flow, concentric tube, shell, and tube heat exchangers perform well due to their geometry.
- Components of refrigeration systems are designed for pure refrigerants; therefore, these designs are not suitable for blended refrigerants.
- It is noted that blended refrigerants can reduce the temperature difference in the heat exchangers due to their non-linearity results as a bigger size of heat exchanger is required.
- Due to the blending of refrigerants the temperature, pressure capacity, and efficiency of the system are changed.

Additional components such as an accumulator and receiver must be added to the circuit for the smooth running of the system as mixed refrigerants can create the problem of choking.

**Author Contributions:** U.S.P. performed the idea development, methodology, and analysis and wrote down the original document. R.S.M. inspected the study and writing for the draft manuscript. R.K.D. corrected the documents for final submission. H.S. prepared the formatting and documentation of

the paper. All authors have understood and agreed to the published edition of the document. All authors have read and agreed to the published version of the manuscript.

**Funding:** This research received no external funding.

**Data Availability Statement:** Data available on request due to restrictions eg privacy or ethical. The data presented in this study are available on request from the corresponding author. The data are not publicly available as research is continue.

**Conflicts of Interest:** The authors declare no conflict of interest.

## Nomenclature

The following nomenclature is used in the manuscript.

| | |
|---|---|
| GWP | Global warming potential |
| ODP | Ozone depletion potential |
| HFO | Hydro fluoro-olefin |
| HC | Hydrocarbon |
| HCFC | Hydro-chlorofluorocarbon |
| CFC | Chlorofluorocarbon |
| TEWI | Total equivalent warming impact |
| COP | Coefficient of performance |
| LLSL-HX | Liquid-line/suction-line heat exchanger |
| $P_1$ | Compressor inlet pressure (bar) |
| $P_2$ | Compressor outlet pressure (bar) |
| $T_1$ | The refrigerant temperature at the inlet of the compressor (°C) |
| $T_2$ | The refrigerant temperature at the outlet of the compressor (°C) |
| $T_3$ | The refrigerant temperature at the inlet of the expansion valve (°C) |
| $T_4$ | The refrigerant temperature at the outlet of the expansion valve (°C) |
| $T_5$ | Water temperature in evaporator(°C) |
| $h_1$ | Enthalpy at the inlet of compressor (KJ/Kg) |
| $h_2$ | Enthalpy at the outlet of compressor (KJ/Kg) |
| $h_3$ | Enthalpy at the inlet of expansion valve (KJ/Kg) |
| $h_4$ | Enthalpy at the outlet of expansion valve (KJ/Kg) |
| Emc | Energy meter constant |
| WD | Work done (KW) |
| RE | Refrigerating effect (KW) |
| $m_w$ | Mass of water in the evaporator unit (litre) |
| $c_p$ | Specific heat at constant pressure |
| $\Delta T$ | Temperature difference (°C) |
| $COP_{actual}$ | Actual coefficient of performance |
| $COP_{theory}$ | Theoretical coefficient of performance |

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
