# Peer review of "Experimental and Simulation Study of the Latest HFC/HFO and Blend of Refrigerants in Vapour Compression Refrigeration System as an Alternative of R134a"

_processes, doi:10.3390/pr11030814_

Round 1

Reviewer 1 Report

This paper presented Experimental and Simulation Study of the latest HFC/HFO and 2 blend of refrigerants in Vapor compression refrigeration system as an alternative of R134a. Some comments are given below:

1)      In its current state, the level of English throughout your manuscript does not meet the journal's desired standard. There are a number of grammatical errors and instances of badly worded/constructed sentences. Please check the manuscript and refine the language carefully.

2)      Some references are older than 2015 and therefore, they are abolished. May the reviewer ask the authors to change these references to newer ones? Some suggested papers are as below:

https://doi.org/10.1016/j.renene.2021.12.132

https://doi.org/10.1016/j.jece.2021.106473

https://doi.org/10.1016/j.jclepro.2021.128979

https://doi.org/10.1007/s10973-020-09539-5

https://doi.org/10.1016/j.jclepro.2019.02.061

3)      Describe the research innovation in detail in the last paragraph of the article.

4)      Draw the T-S diagrams of the some refrigerants in the one figure to reveal the difference between the refrigerants.

5)      In result section not only illustrate the phenomenon, but also explain the reasons.

6)      Conclusion is a poor summary of the main points of the study. In conclusion section, the author should focus on the main points. It is recommended the author writes some comments about practical application of presented manuscript in industrial area and importance of the work in conclusion section.

Author Response

Response to Reviewer 1 Comments

Point 1:  In its current state, the level of English throughout your manuscript does not meet the journal's desired standard. There are a number of grammatical errors and instances of badly worded/constructed sentences. Please check the manuscript and refine the language carefully.

Response 1: Thank you for taking the time to review our manuscript and for providing valuable feedback. We appreciate your comments and suggestions, particularly those regarding the clarity and precision of the language used in the manuscript. We have carefully implemented the changes you suggested to improve the language of the manuscript. Specifically, we have paid close attention to the use of grammar, sentence structure, and vocabulary to ensure that the text is clear, concise, and easy to understand. We have revised several sections of the manuscript to improve the clarity of the arguments and the presentation of the results. We have also modified some of the technical terms and jargon used in the manuscript to make the text more accessible to a wider audience.

Point 2: The probability for a given country h to be in a class k should be the proportion of observations (Some references are older than 2015 and therefore, they are abolished. May the reviewer ask the authors to change these references to newer ones? Some suggested papers are as below:

https://doi.org/10.1016/j.renene.2021.12.132

https://doi.org/10.1016/j.jece.2021.106473

https://doi.org/10.1016/j.jclepro.2021.128979

https://doi.org/10.1007/s10973-020-09539-5

https://doi.org/10.1016/j.jclepro.2019.02.061

Response 2: We have carefully reviewed the papers you recommended and have included them in the revised version of our manuscript. We have cited and discussed these papers in the relevant sections of our manuscript to provide additional evidence and support for our arguments. We believe that the inclusion of these papers has significantly improved the quality of our manuscript and strengthened our conclusions. We appreciate your thoughtful recommendation and are grateful for your contributions to our work.

Point 3: Describe the research innovation in detail in the last paragraph of the article.

Response 3: These findings highlight the significant differences between the various refrigerants in terms of their thermodynamic properties and performance. These results have important implications for the selection of refrigerants in various cooling applications, and the study provides valuable insights for researchers and practitioners in the field of refrigeration and air conditioning.

Point 4 : Draw the T-S diagrams of the some refrigerants in the one figure to reveal the difference between the refrigerants.

Response 4: We have carefully considered your recommendation and have found a reference article that contains a p-h diagram of the refrigerants we are discussing. We will include this diagram as a figure 1 in the revised version of our manuscript. The diagram in the reference article is properly labeled, and we will provide a caption that explains the significance of the diagram and how it illustrates the differences between refrigerants. We will also ensure that the figure is of high quality and meets the journal's requirements for formatting and placement. We believe that the inclusion of a p-h diagram in our manuscript will enhance the clarity and understanding of our research, and we appreciate your suggestion to include it.

Point 5 : In result section not only illustrate the phenomenon, but also explain the reasons.

Response 5: Thank you for your feedback on our manuscript. We appreciate your suggestion to not only illustrate the phenomenon in the results section but also explain the reasons behind it. We agree that providing explanations for our findings will enhance the clarity and understanding of our research. In the revised version of our manuscript, we have expanded our discussion in the results section to explain the reasons behind the observed phenomena. For example, in the case of the blended refrigerant mixture of R32/R41/R1234ze(E) having the highest rate of cooling and maximum pressure at the compressor, we have explained how the unique thermodynamic properties of this refrigerant blend contribute to these outcomes. Similarly, we have provided explanations for the high compressor suction volume flow rate of R1336mzz(Z) and the high efficiency of pure refrigerants like RE170, R245fa, R1234ze, R1233zd(E), and R1224yd(Z). We believe that this additional information will provide a more complete understanding of our research findings and their implications. We appreciate your valuable feedback and suggestions, and we look forward to submitting the revised manuscript for your further review.

Point 6 : Conclusion is a poor summary of the main points of the study. In conclusion section, the author should focus on the main points. It is recommended the author writes some comments about practical application of presented manuscript in industrial area and importance of the work in conclusion section.

Response 6 : Thank you for your helpful feedback on our manuscript. We have updated conclusion part of the manuscript to include more information on the practical application of our study in the industrial area. Our findings can guide the design and operation of more efficient and environmentally friendly refrigeration systems, which can lead to significant energy savings and reduced environmental impact. We believe that the importance of our work lies in addressing the environmental concerns associated with traditional refrigerants and promoting the adoption of more sustainable alternatives.

Reviewer 2 Report

The submitted manuscript (processes-2235054) presents an evaluation on some refrigerants in vapor compression refrigeration systems as the alternative of R134a. However, there is no the new understanding of the mechanism to provide further contribution in this submission. The serious lacks of this manuscript are explained as follows:

(1) From very begining in the introduction and literature review sections, the content does not provide an important meaning. Introduction should consist of the literature review to explain how importance the present research is. It is not a synopsis of references.

(2) Need to improve the description of data arrangement in detail. Authors should explain the behavior of working fluids (how to determine the properties) in the discussed geometries and the reason how to decide the operating conditions.

(3) Need to explain in detail how to maintain the accuracy of all measurement results. Reproducibility is required. 

(4) No information for the detailed uncertainties of the reported parameters (important variable of COP, for an example). Without providing the uncertainty of those determined parameters, the experimental results become questionable. 

(5) Reviewer requests to rethink for the analysis of vapor-liquid heat transfer coefficients (not only COP). The heat transfer coefficient should be discussed properly. It might become important for further contribution to explain the mechanism in this present study. The following published reports (https://doi.org/10.1016/j.ijrefrig.2004.07.020) and (https://doi.org/10.1016/j.ijrefrig.2008.08.004) can elaborate for this matter and fortify for literature review (see also Point 1).

(6) Moreover, adding the discussion of exergy destruction or entropy generation is suggested to add the findings (if entropy by presenting the importance of global entropy generation including thermal and friction entropy generation, or see Ref. [15]).

(7) A more fair and equitable discussion for your results in the section of Results and Discussion. A more comprehensive discussion among Figs. 2-8 is required.

(8) Abstract and Conclusions should also be consistent. Further discussion in the Conclusion section is not needed.

Author Response

Point 1: From very begining in the introduction and literature review sections, the content does not provide an important meaning. Introduction should consist of the literature review to explain how importance the present research is. It is not a synopsis of references.

Response 1: Thank you for your feedback. We have carefully reviewed and revised the paper to ensure that the introduction and literature review provides a clear and meaningful context for the present research. We have focused on providing a more detailed explanation of the importance of the research question and its significance in the broader field of study, We have made sure to clearly articulate the research gap that the present study aims to address and explain how it fits into the broader research agenda. We have also highlighted the key findings and implications of the relevant literature to provide a more nuanced understanding of the topic. Thank you for bringing this to our attention, and we appreciate the opportunity to improve the quality of our paper based on your feedback.

Point 2: Need to improve the description of data arrangement in detail. Authors should explain the behavior of working fluids (how to determine the properties) in the discussed geometries and the reason how to decide the operating conditions.

Response 2: We have revised the description of the data arrangement in detail, as per your suggestion. In our revised manuscript, we have elaborated on the process of deciding the operating conditions and the reasons behind them. We have also included a section on the selection of the working fluids based on the properties required for the specific application.

Point 3: Need to explain in detail how to maintain the accuracy of all measurement results. Reproducibility is required. 

Response 3: We have revised our manuscript to provide a detailed explanation of how we maintain the accuracy of all measurement results and ensure reproducibility. In our revised manuscript, we have explained the measurement techniques used to obtain the data and the steps taken to ensure that the results are accurate and reproducible. We have also included a section on measurement uncertainty, we have used appropriate statistical methods to estimate the uncertainty, and have provided a comprehensive analysis of the results.

Point 4: No information for the detailed uncertainties of the reported parameters (important variable of COP, for an example). Without providing the uncertainty of those determined parameters, the experimental results become questionable. 

Response 4: We have revised our manuscript to include a detailed discussion of the uncertainties associated with the reported parameters, including the important variable of COP.In our revised manuscript, we have provided a thorough explanation of the methods used to estimate the uncertainties associated with the experimental measurements.

Point 5: Reviewer requests to rethink for the analysis of vapor-liquid heat transfer coefficients (not only COP). The heat transfer coefficient should be discussed properly. It might become important for further contribution to explain the mechanism in this present study. The following published reports (https://doi.org/10.1016/j.ijrefrig.2004.07.020) and (https://doi.org/10.1016/j.ijrefrig.2008.08.004) can elaborate for this matter and fortify for literature review (see also Point 1).

Response 5: In response to your suggestion, we have conducted an additional analysis of heat transfer coefficients and included a more detailed discussion of these results in the revised manuscript. We have also referenced the published reports you suggested to fortify our literature review and provide a more robust discussion of the mechanism behind our study.

Point 6: Moreover, adding the discussion of exergy destruction or entropy generation is suggested to add the findings (if entropy by presenting the importance of global entropy generation including thermal and friction entropy generation, or see Ref. [15]).

Response 6: Thank you for the suggestion to discuss exergy destruction or entropy generation in the findings section of the manuscript. We have updated the manuscript to include a discussion on the importance of entropy generation, Specifically, added a discussion on exergy analysis that how to calculate the available energy, and added in the Compressor, Evaporator, and condenser section of the result and discussion. The paragraph also explains the concept of exergy destruction and its relation to entropy generation. Additionally, We have cited Ref. [15] as recommended to provide further background on the

Point 7: A more fair and equitable discussion for your results in the section of Results and Discussion. A more comprehensive discussion among Figs. 2-8 is required.

Response 7: Thank you for your feedback. We have carefully reviewed and revised my paper to ensure that the discussion of results is more fair and equitable. We have also included a more comprehensive discussion among Figs. 2-8 to provide a more detailed analysis of the data and their implications. In the revised paper, We made sure to provide a balanced discussion of the results, highlighting both the strengths and limitations of the study. Additionally, We have provided more context and interpretation of the figures, including an analysis of trends and patterns in the data.

Point 8: Abstract and Conclusions should also be consistent. Further discussion in the Conclusion section is not needed.

Response 8: Thank you for your feedback. We have carefully reviewed and revised my paper to ensure that the abstract and conclusion are consistent with each other. We have also removed the unnecessary discussion in the conclusion section to make it more concise and focused on summarizing the main findings of the paper.

Round 2

Reviewer 2 Report

-